# EpiFormer: A Transformer-Based Multi-Relational Equivariant Graph Neural Network for Antibody-Aware Epitope Prediction

## Abstract

Antibodies are essential components of the immune system, neutralizing foreign antigens such as viruses by binding to specific regions called epitopes. Computational prediction of epitopes is critical for antibody design and therapeutic development. Current approaches for epitope prediction still remain challenging due to: (1) lack of sophisticated architectures to model the complex interaction patterns; (2) ineffective protein representations; (3) antibody-agnostic modeling despite antibody specificity; (4) severe class imbalance; and (5) scarcity of known antigen–antibody complexes. In order to overcome these challenges, we propose *EpiFormer*, an encoder-decoder-based architecture that utilizes an E(3)-equivariant multi-relational graph neural network (GNN) coupled with cross-attention to model antigen-antibody interactions. Our contributions are an E(3)-equivariant multi-relational GNN, a Transformer-style cross-attention mechanism, and tailored losses for severe class imbalance and data scarcity. Our method significantly outperforms existing baselines on the Antibody-specific Epitope Prediction (AsEP) dataset by achieving an overall $\approx 1.7x$ performance improvement on multiple classification metrics. This work advances the state-of-the-art in antibody-aware epitope prediction, providing a robust framework for therapeutic antibody design and vaccine development.

## 1 Introduction

Antibodies are large, Y-shaped proteins produced by B-cells that play a critical role in the immune system by identifying and neutralizing foreign substances such as toxins, bacteria, and viruses, collectively known as antigens. They are currently known to be the largest class of biotherapeutics, where five of the current top 10 blockbuster drugs are monoclonal antibodies (Norman et al., 2020; Joubbi et al., 2024). Recently, computational approaches have been proposed to design antibodies to aid the existing traditional approaches that are time-consuming, expensive, and laborious (Fischman & Ofran, 2018; Krishnan et al., 2024; Hummer et al., 2022). An important step in computational antibody design is antigen binding site or epitope prediction, which involves identifying the residues on the surface of an antigen that are recognized and bound by an antibody (Zeng et al., 2023). Accurate epitope prediction is also essential for understanding antibody-antigen interactions in biomedical research (Krishnan et al., 2024).

Despite significant advances in deep learning-based protein binding site prediction methods, current approaches for epitope prediction encounter limitations that severely constrain their effectiveness (Wang et al., 2024a; Fang et al., 2023). (1) Existing architectures lack the sophistication to model complex interaction patterns, with standard GNN struggling to differentiate and learn the distinct geometries of antigens and antibodies while missing essential 3D-related inductive biases like translational invariance and rotational/reflectional equivariance (Zhang et al., 2022). 2) Most methods rely on ineffective protein representations, predominantly using sequence-based approaches that fail to capture the complex three-dimensional spatial arrangements of antigen binding sites, despite epitopes being inherently non-linear and conformationally diverse (Hummer et al., 2022). 3) Most approaches remain agnostic to pre-conditioned antibodies and treat epitope prediction as an antibody-independent problem despite the fact that epitopes are antibody-specific (Norman et al., 2020). 4) The epitope prediction problem suffers from severe data imbalance, as epitopic residues

comprise merely 10% of all residues in an antigen (Liu et al., 2024). 5) The sparsity of known antigen-antibody complexes creates a fundamental data limitation, with only approximately 2,000 available interaction pairs compared to millions of general protein structures (Joubbi et al., 2024).

To address these fundamental challenges, we propose *EpiFormer*, an encoder-decoder architecture that utilizes E(3)-equivariant graph neural networks (EGNN) on multi-relational protein graphs coupled with cross-attention mechanisms to model antibody-antigen interactions. We represent antigens and antibodies as multi-relational graphs and introduce essential 3D-related inductive biases, including translational invariance and rotational equivariance via equivariant message passing to effectively handle protein geometric constraints that standard GNNs cannot capture (Liao & Smidt, 2022). We train *EpiFormer* with a custom joint objective that addresses the severe epitopic class imbalance, predicts antigen-antibody interaction maps, and enforces geometric consistency through inter-chain distance constraints. The framework operates in an antibody-aware manner by explicitly incorporating antibody structure and binding context through bidirectional cross-attention mechanisms that enable dynamic modeling of both intra-chain geometric relationships and inter-chain interaction patterns (Lim et al., 2025). Our main contributions are as follows:

1. We develop a novel transformer-based GNN architecture that achieves $\approx 1.7$x performance improvement over existing baselines on the antibody-aware epitope prediction task on multiple classification metrics.

2. We introduce a multi-relational E(3)-equivariant message passing (EGNN-R) framework that handles multiple edge relations for robust epitope prediction.

3. We develop *EpiFormer* with a novel joint loss function designed for: (a) severely imbalanced epitopic data, (b) interaction map prediction, and (c) inter-chain geometric consistency.

## 2 RELATED WORK

GNN have emerged as a powerful approach for epitope prediction by modeling the spatial and sequential relationships in protein structures. Several methods demonstrate this approach: *PECAN* (Pittala & Bailey-Kellogg, 2020), *PInet* (Dai & Bailey-Kellogg, 2021), and related work (Jha et al., 2022) use GNNs with attention mechanisms for protein-protein interaction prediction. *EPMP* (Vecchio et al., 2021) uses a neural message-passing framework with asymmetrical architectures for paratope-epitope prediction. Recent advances combine protein language model (PLM) embeddings with graph-based architectures, with *EpiGraph* (Choi & Kim, 2024) using GAT with ESM-2 embeddings, *AsEP* (Liu et al., 2024) employing the *WALLE* method with ESM-2 and AntiBERTy embeddings, and *GraphBepi* (Zeng et al., 2023) leveraging ESM-2 representations.

These graph-based methods can be categorized based on whether they use antibody-specific information. Antibody-agnostic approaches, such as *epitope1D* (Silva et al., 2023), *GraphBepi* (Zeng et al., 2023), and *EpiGraph* (Choi & Kim, 2024), rely on sequential and structural features but lack specificity for antibody-specific applications (Vecchio et al., 2021). In contrast, antibody-aware methods like *EpiScan* (Wang et al., 2024a), *PECAN* (Pittala & Bailey-Kellogg, 2020), and *EPMP* (Vecchio et al., 2021) explicitly incorporate antibody structure or sequence information. Some approaches like (Lu et al., 2022) combine GNNs with attention-based bidirectional LSTM networks to capture both local spatial information and global sequence information from antigens.

EGNN have gained attention for protein structure modeling because they preserve geometric properties under rotations and translations, which are essential for capturing 3D protein conformations (Satorras et al., 2021b; Schütt et al., 2018). Traditional GNNs often fail to maintain these geometric constraints when processing protein structures, leading to suboptimal representations of spatial relationships. E(3)-equivariant approaches like EGNN (Satorras et al., 2021b) and GearNet (Zhang et al., 2022) address this limitation by incorporating equivariance directly into the message-passing framework. Multi-relational graphs further improve protein modeling by representing different types of interactions through distinct edge types (Zhang et al., 2022). Recent work has applied these concepts to protein-protein interactions, with methods like (Liao & Smidt, 2022) using equivariant transformers for molecular modeling and (Lim et al., 2025) employing multi-relational representations for protein-ligand binding affinity prediction.

## 3 METHODS

In this section, we present the graph construction, problem formulation, and the architecture of *EpiFormer*, a model designed for antibody-aware epitope prediction. *EpiFormer* takes as input an antigen and an antibody, and predicts their binding sites by dynamically modeling their interaction using geometric message passing and cross-attention. We then present the customized loss functions tailored for antigen-antibody interaction prediction to train *EpiFormer*.

### 3.1 PRELIMINARIES

**Graph construction** The protein 3D structure is described as a point cloud of atoms $\{v_{i,k}\}_{1 \leq i \leq p, 1 \leq k \leq p_i}$, where $p_i$ is the number of atoms in residue $v_i$ and $p$ represents the number of amino acid residues in the protein. The first four atoms in any residue correspond to its backbone atoms (N, $C_\alpha$, $C_\beta$, O) and the rest are its side chain atoms. The 3D coordinate of an atom $v_{i,k}$ is denoted as $x(v_{i,k}) \in \mathbb{R}^3$. Since we work with the *unbound* structures or point clouds of antigen(ag) and antibody(ab), we build two completely independent residue graphs $\mathcal{G}_{ag} = (\mathcal{V}_{ag}, \mathcal{E}_{ag}, \mathcal{R})$, and $\mathcal{G}_{ab} = (\mathcal{V}_{ab}, \mathcal{E}_{ab}, \mathcal{R})$. Vertex $v_i \in \mathcal{V}$ represents residue $i$, centered on $C_\alpha$ at coordinate $\mathbf{x}_i \in \mathbb{R}^3$. $|\mathcal{V}_{ag}| = n$, $|\mathcal{V}_{ab}| = m$, and edges $e_{i,j} \in \mathcal{E}$ encode structural/functional relationships between residues.

Each node $v_i \in \mathcal{V}$ is attributed a node feature vector $\mathbf{h}_i \in \mathbb{R}^{d_h}$ and a node coordinate matrix $\mathbf{X}_i \in \mathbb{R}^{3 \times 4}$ consisting of four backbone atoms $\xi = \{N, C_\alpha, C_\beta, O\}$ ($\mathbf{x}_i$ is short for $\mathbf{x}_{i,C_\alpha}$). Specifically, the node feature vector $\mathbf{h}_i$ constitutes handcrafted geometric features and PLM-derived embeddings to capture both structural and evolutionary information. In addition, each edge $e_{i,j}$ is attributed an edge feature vector $\mathbf{f}_{i,j} \in \mathbb{R}^{d_f}$ and a tuple of edge relations $\mathbf{r}_{i,j} \subseteq \mathcal{R}$. The edge vector $\mathbf{f}_{i,j}$ encodes features such as distances and angles to capture both local geometry and global structural context. The set of edge relations $\mathcal{R} = \{\rho_1, \rho_2, \rho_3, \rho_4\}$ captures distinct protein interactions: sequential relations for peptide bonds ($\rho_1$) and short-range coupling ($\rho_2$), plus spatial relations for local packing shells via $K$-nearest neighbors ($\rho_3$) and medium-range contacts within 8 Å ($\rho_4$). Please refer to Appendix A.3 for further details. We extend the notation of these attributes to refer to the residue graph $\mathcal{G}$ of the antigen (or antibody) as $(\mathbf{H}, \mathbf{X}, \mathbf{F}, \mathbf{R})$.

**Problem Formulation** We formulate the problem as the following two tasks:

*Epitope node prediction:* A binary node classification task where a residue $v \in \mathcal{V}_{ag}$ is labeled as an epitope (1) if it is within 4.5Å of any residue in $\mathcal{V}_{ab}$; otherwise, it is labeled as a non-epitope (0). The classifier predicts the epitope node labels $\hat{y}_{ag}$ using $f : v_{ag} \to \{0, 1\}$ and is defined as:

$$\hat{y}_{ag} = f(v_{ag}; \mathcal{G}_{ag}, \mathcal{G}_{ab}) = \begin{cases} 1 & \text{if } v_{ag} \text{ is an epitope,} \\ 0 & \text{otherwise.} \end{cases} \quad (1)$$

*Bipartite graph link prediction:* This task predicts the bipartite adjacency matrix $\hat{\mathcal{E}}_{bg}$ between antibody and antigen in the bipartite graph $\mathcal{G}_{bg} = (\mathcal{V}_{ag} \cup \mathcal{V}_{ab}, \mathcal{E}_{bg})$, where $\mathcal{V}_{ag}$ and $\mathcal{V}_{ab}$ are disjoint vertex sets, and $\mathcal{E}_{bg} \subseteq \mathcal{V}_{ag} \times \mathcal{V}_{ab} \in \{0, 1\}^{n \times m}$ denotes inter-molecular contacts between antigen and antibody. An edge $e_{bg} \in \mathcal{E}_{bg}$ is a contact (labeled as 1) if the corresponding residues $(v_{ag}, v_{ab})$ are within 4.5Å of each other and 0 otherwise. The edge classifier $g : e_{bg} \to \{0, 1\}$ is defined as:

$$\hat{\mathcal{E}}_{bg} = g(e_{bg}; \mathcal{G}_{bg}) = \begin{cases} 1 & \text{if } e_{bg} \text{ is a contact,} \\ 0 & \text{otherwise.} \end{cases} \quad (2)$$

**Equivariance and Invariance in E(3) Space** Traditional graph representations of proteins capture connectivity but ignore crucial 3D geometric information. Recently, proteins have been naturally modeled as geometric graphs that encode both topological connectivity and 3D spatial coordinates of atoms. Since molecular properties remain unchanged under rigid body transformations (rotations, translations, reflections), geometric GNN incorporate E(3)-equivariance as an inductive bias to respect these fundamental symmetries (Jiao et al., 2023).

For a protein with coordinates $\mathbf{X} \in \mathbb{R}^{3 \times m}$ and scalar features $\mathbf{h} \in \mathbb{R}^d$, an E(3)-equivariant function $f$ satisfies:

$$f(g \cdot \mathbf{X}, \mathbf{h}) = g \cdot f(\mathbf{X}, \mathbf{h}), \quad \forall g \in \text{E}(3) \quad (3)$$

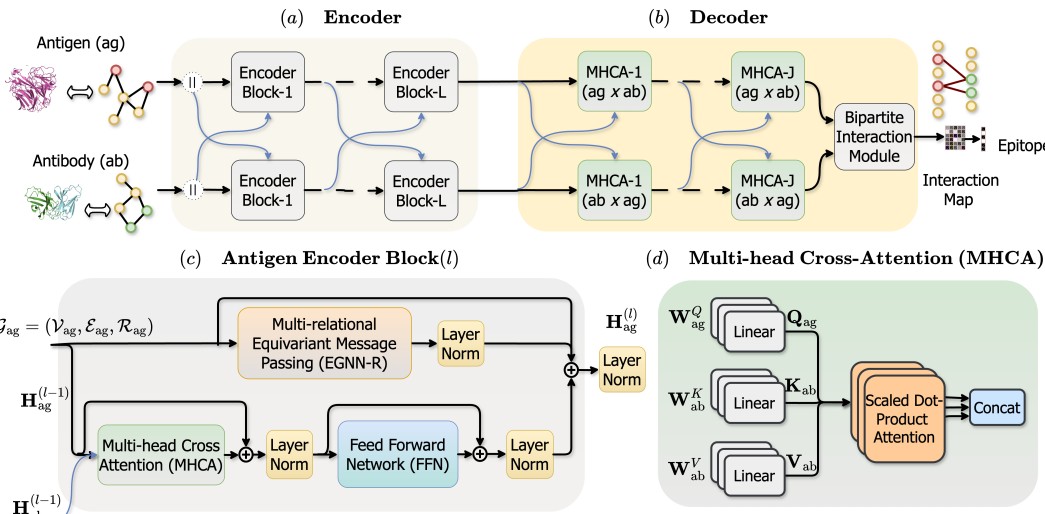

Figure 1: Overview of *EpiFormer*. The inputs are an antigen multi-relational graph $\mathcal{G}_{ag} = (\mathcal{V}_{ag}, \mathcal{E}_{ag}, \mathcal{R})$ and an antibody multi-relational graph $\mathcal{G}_{ab} = (\mathcal{V}_{ab}, \mathcal{E}_{ab}, \mathcal{R})$, while the outputs are the bipartite adjacency matrix and the binary epitope node labels. (a) Antigen and antibody graphs are encoded with parallel multi-relational equivariant message passing layers (EGNN-R) and cross-attention blocks. "||" is a small gating network determines the relative importance of geometric and language features for every residue. (b) A bi-directional cross-attention decoder produces the interaction map. (c) Antigen Encoder Block schematic (Antibody Encoder Block is analogous) where "⊕" denotes addition. (d) An example of MHCA between antigen and antibody.

where group actions are defined as translations $g \cdot \mathbf{X} = \mathbf{X} + \mathbf{b}$ or rotations/reflections $g \cdot \mathbf{X} = \mathbf{OX}$ with $\mathbf{O} \in O(3)$. This is contrast to E(3)-invariant functions, which satisfy $f(g \cdot \mathbf{X}, \mathbf{h}) = f(\mathbf{X}, \mathbf{h})$, producing outputs unchanged by coordinate transformations.

## 3.2 EPIFORMER

In this section, we present the architecture of *EpiFormer*, an encoder-decoder framework for antibody-antigen binding-site prediction. The model receives two disjoint multi-relational residue graphs, $\mathcal{G}_{ag}$ and $\mathcal{G}_{ab}$, processes them with independent encoders that produces residue-level embeddings, and then passes these embeddings to a decoder to reconstruct the bipartite adjacency matrix $\hat{\mathcal{E}}_{bg} \in \{0, 1\}^{n \times m}$. A desirable property of our proposed framework is its E(3)-equivariance to address a broader range of symmetries in antigen-antibody interactions and preserve the geometry of these proteins. The overall workflow is presented in Figure 1 while the algorithm is provided in the Appendix 1.

**Encoder** The *EpiFormer* contains two parallel encoders with no shared parameters, one dedicated to the antigen chain and the other to the antibody chain, as shown in Figure 1 **(a)**. Both encoders operate on heterogeneous residue graphs $\mathcal{G}_{ag}$ and $\mathcal{G}_{ab}$ whose nodes encode Cartesian coordinates $\mathbf{x}_i \in \mathbb{R}^3$, geometric descriptors $\mathbf{h}_i^{geo} \in \mathbb{R}^{d_{geo}}$ and PLM embeddings $\mathbf{h}_i^{plm} \in \mathbb{R}^{d_{plm}}$. Before message passing begins, a small gating network determines the relative importance of geometric and language features for every residue. The gate first concatenates the two feature vectors, applies a linear projection, and normalises the result with a softmax, $g_i = \text{Softmax}(\mathbf{W}_g[\mathbf{h}_i^{geo} \| \mathbf{h}_i^{plm}])$, where $\mathbf{W}_g \in \mathbb{R}^{2 \times d_h}$ is the weight matrix of the gate network with $d_h = d_{geo} + d_{plm}$. It then combines the inputs through feature-specific projections to the working width $d_h$:

$$\mathbf{h}_i^0 = \sum_{k \in \{geo, plm\}} g_{ik} \mathbf{W}_k \mathbf{h}_i^{(k)} \in \mathbb{R}^{d_h}. \tag{4}$$

The vector $\mathbf{h}_i^0$ serves as the initial node state for the first *EpiFormer* encoder block. The schematic of an *EpiFormer* block is shown in Figure 1 **(c)**. Let $\mathbf{H}_{ag}^\ell \in \mathbb{R}^{n \times d_h}$ and $\mathbf{H}_{ab}^\ell \in \mathbb{R}^{m \times d_h}$ be the current embeddings, which are passed in parallel to their EGNN-R and MHCA layers.

*Relation-aware EGNN (EGNN-R) layer*: We develop a relation-aware variant of EGNN (Satorras et al., 2021a) to propagate structural and geometric information within each chain. Let $\mathbf{h}_i^\ell \in \mathbb{R}^{d_h}$ and $\mathbf{x}_i^\ell \in \mathbb{R}^3$ denote the feature and coordinate of residue $i$ after the $\ell$-th EGNN-R layer. Every undirected edge $e_{i,j}$ carries a tuple $\mathbf{r}_{i,j} \subseteq \mathcal{R}$ that encodes sequential and spatial relations. With the squared distance $d_{ij} = \|\mathbf{x}_i^\ell - \mathbf{x}_j^\ell\|_2^2$ and the displacement vector $\boldsymbol{\delta}_{ij} = \mathbf{x}_i^\ell - \mathbf{x}_j^\ell$, the layer performs the following computations:

$$m_{ij}^\rho = \phi_m^\rho(\mathbf{h}_i^\ell, \mathbf{h}_j^\ell, \gamma(d_{ij}), \mathbf{f}_{ij}), \qquad \mathbf{h}_i^{(\ell+1)} = \mathbf{h}_i^\ell + \phi_h\Big(\mathbf{h}_i^\ell, \sum_{j \in \mathcal{N}(i)} \sum_{\rho \in \mathbf{r}_{ij}} m_{ij}^\rho\Big), \quad (5)$$

$$s_{ij}^\rho = \phi_x^\rho(m_{ij}^\rho), \qquad \mathbf{x}_i^{(\ell+1)} = \mathbf{x}_i^\ell + \sum_{j \in \mathcal{N}(i)} \sum_{\rho \in \mathbf{r}_{ij}} \frac{\boldsymbol{\delta}_{ij}}{\sqrt{d_{ij} + \varepsilon}}\, s_{ij}^\rho. \quad (6)$$

Here, $\gamma(\cdot)$ denotes a 16-term radial basis function, $\mathbf{f}_{ij}$ is the edge's attribute vector, and each mapping $\phi_{\{m,x\}}^\rho$ is realized as a two-layer multilayer perceptron whose parameters are shared by all edges with the same relation label $\rho$, and $\varepsilon = 10^{-8}$. Specifically, we have four relation-specific message MLPs $\phi_m^\rho : \mathbb{R}^{2d_h + d_f + 16} \to \mathbb{R}^{d_q}$ and coordinate MLPs $\phi_x^\rho : \mathbb{R}^{d_q} \to \mathbb{R}^3$, and a node update MLP $\phi_h : \mathbb{R}^{d_h + d_q} \to \mathbb{R}^{d_h}$ shared across all relations, where $d_q$ represents hidden layer dimension. Applying residual connections and layer normalization produces output embeddings at layer $\ell$ as:

$$\mathbf{H}_{\text{ag}}^{\text{intra}} = \{\, W_{\text{ag}}^\ell \mathbf{h}_i^\ell \mid v_i \in \mathcal{V}_{\text{ag}}\}, \qquad \mathbf{H}_{\text{ab}}^{\text{intra}} = \{\, W_{\text{ab}}^\ell \mathbf{h}_j^\ell \mid v_j \in \mathcal{V}_{\text{ab}}\}, \quad (7)$$

where $W^\ell$ represents the trainable parameters for the EGNN-R layer $\ell$ for each *EpiFormer* encoder block and $\mathbf{H}_{\{\text{ag,ab}\}}^{\text{intra}}$ represents the output residue embeddings of antigen and antibody after passing through their respective EGNN-R layer $\ell$. The layer remains E(3)-equivariant by construction because the only vector quantity entering the coordinate update is the displacement $\boldsymbol{\delta}_{ij}$, while cross-attention works with rotation and translation-invariant features (Liao & Smidt, 2022) (please refer to Appendix A.1 for the formal proof).

*Multi-head cross-attention (MHCA) layer with feed-forward network*: In parallel to geometric message passing, each encoder block applies bidirectional multi-head cross-attention (MHCA) (Vaswani et al., 2017) to enable inter-chain communication. The MHCA mechanism shown in Figure 1 **(d)**, produces cross-chain context representations $\widetilde{\mathbf{H}}_{\text{ag}}$ and $\widetilde{\mathbf{H}}_{\text{ab}}$. A learnable scalar gate $\alpha$ balances intra-chain geometry with cross-chain context:

$$\mathbf{H}_{\text{ag}}^{(\ell+1)} = \mathbf{H}_{\text{ag}}^\ell + \mathbf{H}_{\text{ag}}^{\text{intra}} + \alpha_{\text{ag}}\, \text{FFN}(\widetilde{\mathbf{H}}_{\text{ag}}), \quad \mathbf{H}_{\text{ab}}^{(\ell+1)} = \mathbf{H}_{\text{ab}}^\ell + \mathbf{H}_{\text{ab}}^{\text{intra}} + \alpha_{\text{ab}}\, \text{FFN}(\widetilde{\mathbf{H}}_{\text{ab}}), \quad (8)$$

where $\alpha_{\text{ag}}, \alpha_{\text{ab}} \in \mathbb{R}^+$ are learnable parameters, $\widetilde{\mathbf{H}} = \text{MHCA}(\mathbf{H})$, and FFN is a two-layer Feed Forward Network. The MHCA is detailed in Appendix A.2.

**Decoder** The decoder refines the residue embeddings $\mathbf{H}_{\text{ag}}^L$ and $\mathbf{H}_{\text{ab}}^L$ and performs bipartite interaction prediction. The decoder has $J$ identical layers, each containing: (i) bidirectional MHCA with FFN, and (ii) layer normalization with residual connections, followed by a bipartite interaction head. The embeddings $\mathbf{H}_{\text{ag}}^J$ and $\mathbf{H}_{\text{ab}}^J$ serve as inputs to the bipartite interaction module.

*Bipartite interaction prediction module* The bipartite adjacency matrix is obtained by projecting the embeddings into queries and keys of width $d_k$ in both directions, forming scaled dot-product similarities:

$$\mathbf{S}_{\text{ag} \to \text{ab}} = \frac{(\mathbf{H}_{\text{ag}}^L \mathbf{W}_Q^{\text{out}})(\mathbf{H}_{\text{ab}}^L \mathbf{W}_K^{\text{out}})^\top}{\sqrt{d_k}}, \qquad \mathbf{S}_{\text{ab} \to \text{ag}} = \frac{(\mathbf{H}_{\text{ab}}^L \mathbf{W}_Q'^{\text{out}})(\mathbf{H}_{\text{ag}}^L \mathbf{W}_K'^{\text{out}})^\top}{\sqrt{d_k}}. \quad (9)$$

The two score maps are fused via a learnable mixing vector $\mathbf{w} \in \mathbb{R}^2$ and bias $b \in \mathbb{R}$ to produce logits $\mathbf{Z} = \mathbf{w}^\top [\, \mathbf{S}_{\text{ag} \to \text{ab}}\, (\mathbf{S}_{\text{ab} \to \text{ag}})^\top\, ] + b$, and the interaction probabilities are $\hat{\mathcal{E}}_{\text{bg}} = \sigma(\mathbf{Z}) \in \mathbb{R}^{n \times m}$.

## 3.3 JOINT OBJECTIVE

*EpiFormer* is trained with a joint objective that consists of a bipartite edge reconstruction loss, epitope node classification loss, and an auxiliary inter-chain distance classification objective. The overall

training objective is a weighted sum of these loss components:

$$\mathcal{L} = \lambda_{\text{edge}} \, \mathcal{L}_{\text{edge}} + \lambda_{\text{node}} \, \mathcal{L}_{\text{node}} + \lambda_{\text{geo}} \, \mathcal{L}_{\text{geo}}. \tag{10}$$

**Edge Prediction Loss** ($\mathcal{L}_{\text{edge}}$)  This loss applies positive-class-reweighted binary cross-entropy over all antigen-antibody residue pairs:

$$\mathcal{L}_{\text{edge}} = -\frac{1}{nm} \sum_{i=1}^{n} \sum_{j=1}^{m} \left[ \pi_{\text{edge}} \, (\mathcal{E}_{\text{bg}})_{ij} \log(\hat{\mathcal{E}}_{\text{bg}})_{ij} + (1 - (\mathcal{E}_{\text{bg}})_{ij}) \log(1 - (\hat{\mathcal{E}}_{\text{bg}})_{ij}) \right], \tag{11}$$

where $\mathcal{E}_{\text{bg}} \in \{0,1\}^{n \times m}$ is the ground-truth interaction matrix per complex, and $\pi_{\text{edge}}$ compensates for the extreme sparsity of positives. This loss directly supervises the bipartite interaction prediction, which serves as the foundation for deriving epitope probabilities.

**Node Classification Loss** ($\mathcal{L}_{\text{node}}$)  The node classification loss supervises epitope nodes only and combines three complementary objectives to handle class imbalance and enforce structural priors:

$$\mathcal{L}_{\text{node}} = \beta_{\text{BCE}} \, \mathcal{L}_{\text{BCE}}^{\text{epi}} + \beta_{\text{Dice}} \, \mathcal{L}_{\text{Dice}}^{\text{epi}} + \beta_{\text{sparsity}} \, \mathcal{L}_{\text{sparsity}}^{\text{epi}}, \tag{12}$$

where $\beta_{\{\cdot\}}$ weight the different terms. The probability that node $v_{\text{ag}}$ is an epitope is derived from the bipartite interaction matrix using a top-$k$ pooling strategy which captures the relationship between $v_{\text{ag}}$ and nodes of the antibody :

$$(\hat{y}_{\text{ag}})_i = \frac{1}{k} \sum_{j \in \text{top-}k(\hat{\mathcal{E}}_{\text{bg}})_{i:}} (\hat{\mathcal{E}}_{\text{bg}})_{ij}, \tag{13}$$

where $(\hat{\mathcal{E}}_{\text{bg}})_{i:}$ denotes the $i$-th row, and $k$ is determined using cross-validation.

*Class-Reweighted Binary Cross-Entropy:* The primary classification loss applies positive class reweighting (with $\pi_{\text{epi}} > 1$) to address the severe class imbalance in epitope prediction:

$$\mathcal{L}_{\text{BCE}}^{\text{epi}} = -\frac{1}{n} \sum_{i=1}^{n} [\pi_{\text{epi}} \, (y_{\text{ag}})_i \log(\hat{y}_{\text{ag}})_i + (1 - (y_{\text{ag}})_i) \log(1 - (\hat{y}_{\text{ag}})_i)]. \tag{14}$$

*Dice Loss for Graph Segmentation:* The Dice loss treats epitope prediction as a segmentation problem which is effective for highly imbalanced image segmentation (Sudre et al., 2017):

$$\mathcal{L}_{\text{Dice}}^{\text{epi}} = 1 - \frac{2 \sum_{i=1}^{n} (\hat{y}_{\text{ag}})_i (y_{\text{ag}})_i + \alpha}{\sum_{i=1}^{n} (\hat{y}_{\text{ag}})_i + \sum_{i=1}^{n} (y_{\text{ag}})_i + \alpha}, \tag{15}$$

where $\alpha > 0$ is a small smoothing constant for numerical stability. The Dice coefficient measures the overlap between predicted and true epitope regions, with the loss being $1 - \text{Dice}$.

*Per-Graph Sparsity Regularization:* The sparsity term enforces cardinality matching between predicted and true epitope counts for each complex in the mini-batch:

$$\mathcal{L}_{\text{sparsity}}^{\text{epi}} = ||\hat{y}_{\text{ag}} - y_{\text{ag}}||_1. \tag{16}$$

This regularizer is crucial for calibrating predictions across complexes of varying sizes.

**Auxiliary Distance Classification Loss** ($\mathcal{L}_{\text{geo}}$)  The auxiliary geometric term provides additional supervision by classifying inter-chain distances into discrete bins, helping the model learn geometrically meaningful representations. The loss focuses on near-contact pairs and ignores distant residue pairs that are unlikely to interact. This auxiliary supervision encourages the model to learn distance-aware representations while still maintaining focus on the primary epitope prediction task.

Let $\mathcal{M} = \{(i,j) : d_{ij} \leq D_{\text{max}}\}$ be the set of antigen-antibody residue pairs within the maximum distance cutoff, where $d_{ij}$ is the Euclidean distance between residues $i$ and $j$. The bins are defined by distances $\{d_0, d_1, d_2, d_3, d_4\} = \{0, 4, 8, 16, 32\}$ Å, creating $B = 4$ bins:

$$b(i,j) = \arg \max_{b \in \{1,\ldots,4\}} \mathbf{1}[ \, d_{b-1} \leq d_{ij} < d_b ]. \tag{17}$$

The network predicts per-pair distance logits $\boldsymbol{\Delta}_{ij} \in \mathbb{R}^5$, but only the first $B = 4$ components $\widehat{\boldsymbol{\Delta}}_{ij} \in \mathbb{R}^4$ are used for pairs in $\mathcal{M}$, ignoring the "far" class beyond $D_{\max} = 32\,\text{Å}$. The class probabilities are:

$$p_{ijb} = \frac{\exp(\widehat{\Delta}_{ijb})}{\sum_{b'=1}^{4} \exp(\widehat{\Delta}_{ijb'})}. \tag{18}$$

The loss combines class balancing with distance-aware weighting:

$$\mathcal{L}_{\text{geo}} = -\frac{1}{|\mathcal{M}|} \sum_{(i,j) \in \mathcal{M}} w_{ij} \sum_{b=1}^{4} \alpha_b \, \mathbf{1}[\, b(i,j) = b \,] \log p_{ijb}, \tag{19}$$

where $\alpha_b > 0$ are class-balance weights computed from empirical bin frequencies within $\mathcal{M}$ and $w_{ij} > 0$ are distance weights inversely proportional to $d_{ij}$, normalized to unit mean over $\mathcal{M}$.

## 4 EXPERIMENTS AND RESULTS

**Dataset** We utilized the AsEP dataset (Liu et al., 2024), a novel benchmark dataset of antibody-antigen complexes designed specifically for epitope prediction tasks. After preprocessing, we retain 1,723 unique antibody–antigen complexes; details are in Appendix A.3.4. We excluded two complexes (5nj6_0P and 5ies_0P) from the AsEP dataset due to sequence alignment inconsistencies and unresolved residues, with the final dataset containing 1,721 complexes.

Our EDA revealed several key insights into the dataset and is shown in Figure 2. The distribution of epitope residues showed a mean of 19 ± 4.7, while the antigen surface residues numbered in the hundreds. The contact distribution between residues in the bipartite graph had a mean of 43.7 contacts with a standard deviation of 12.8. Additionally, the dataset includes 641 unique antigens and 973 epitope groups, highlighting the diversity and complexity of the antibody-antigen interactions captured in the AsEP dataset.

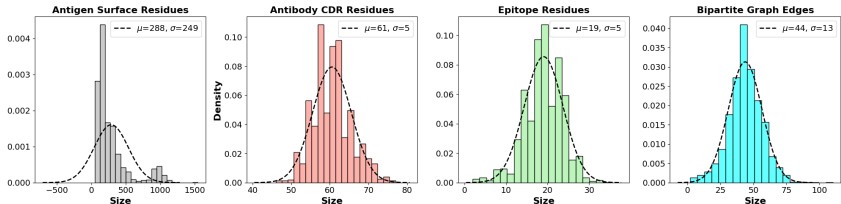

Figure 2: The size distribution of the antigen surface residues, antibody CDR residues, epitope residues, and antibody-antigen bipartite graph edges in the AsEP dataset.

*Stratified Splits:* We adopt two stratified splitting strategies from the AsEP benchmark dataset (Liu et al., 2024): epitope-to-antigen surface ratio split and epitope-group split. The first approach stratifies complexes by the ratio ($\#epitope\_nodes/\#antigen\_nodes$) to balance the class imbalance between interface and non-interface residues across train, validation, and test sets. Given that epitopes are typically limited in size (approximately $14.6 \pm 4.9$ residues) whereas antigen surfaces often contain several hundred residues, this stratification controls task difficulty by matching the distribution of epitope-to-surface ratios across splits.

The epitope-group split employs a different strategy by clustering complexes by antigen epitope and completely excluding test epitopes from training and validation data to evaluate model performance on novel binding sites. The dataset also includes multi-epitope antigens for which different antibodies bind distinct locations on the same antigen, and the split follows an 80/10/10 allocation by complexes. Both dataset splits result in 1,381 training complexes and 170 complexes each for validation and testing.

### 4.1 BASELINE COMPARISON

We evaluate the model performance using standard classification metrics such as Matthews Correlation Coefficient (MCC), Area Under the Receiver Operating Characteristic Curve (AUC-ROC), Area

Under the Precision-Recall Curve (AUPRC), accuracy, precision, recall, and F1 score. Table 1 presents the performance comparison of *EpiFormer* with the existing baseline methods for epitope prediction. We trained several baseline methods for epitope prediction on the AsEP dataset and report their results to establish a fair comparison with our model using their reported training configurations. Methods such as WALLE (Liu et al., 2024), MIPE (Wang et al., 2024b), and EpiScan (Wang et al., 2024a) are antibody-aware while others such as EpiGraph (Choi & Kim, 2024) are not. *EpiFormer* achieves the strongest overall performance, with best AUC/AUPRC/F1/MCC across both evaluation settings (epitope-ratio and epitope-group splits), outperforming these antibody-agnostic and antibody-aware epitope prediction methods. It can also be seen that, unlike our method, the existing baselines do not provide consistent overall performance on the classification metrics. We also evaluated *EpiFormer* on the challenging epitope-group split and achieved F1/MCC scores of 0.228/0.168, compared to next best-performing baseline model, WALLE, which achieved F1/MCC scores of 0.145/0.077. This performance gap highlights the importance of modeling both geometric constraints and dynamic antibody-antigen interactions for accurate epitope prediction. The table 2 summarizes whether each baseline conditions on antibody inputs, uses structural topology, leverages PLM representations, and adopts a graph representation, and additionally whether it incorporates explicit geometric surface/3D features, models multi-relational edges, employs equivariant GNNs, or cross-attention.

Table 1: Performance comparison of epitope prediction baseline methods with *EpiFormer* on the AsEP dataset using the epitope-to-surface ratio stratified split. The best values are represented in bold, while the second-best values are underlined.

| Method | AUC | AUPRC | F1 | MCC | Precision | Recall |
|---|---|---|---|---|---|---|
| EpiGraph | 0.819 | 0.279 | 0.247 | 0.240 | 0.145 | 0.852 |
| EpiScan | 0.593 | 0.229 | 0.197 | 0.043 | 0.115 | **0.912** |
| MIPE | 0.774 | 0.213 | 0.169 | 0.176 | 0.317 | 0.248 |
| WALLE | 0.635 | 0.2195 | 0.258 | 0.210 | 0.235 | 0.422 |
| **EpiFormer (ours)** | **0.889** | **0.443** | **0.433** | **0.404** | **0.329** | 0.633 |

Table 2: Summary of features and modeling choices in baseline methods. Antibody: uses antibody information for epitope prediction; Structure: uses structure/topology as model input; PLM: uses pre-trained PLM embeddings; Graph: uses a graph representation; Geom.: explicit geometric surface/3D features; Multi-rel.: uses relation-aware/multi-edge types; E(3)-Eq.: uses an E(3)-equivariant GNN; Cross-Attn.: employs cross-attention between antibody and antigen representations.

| Method | Antibody | Structure | PLM | Graph | Geom. | Multi-rel. | E(3)-Eq. | Cross-Attn. |
|---|---|---|---|---|---|---|---|---|
| EpiGraph | ✗ | ✓ | ✓ | ✓ | ✓ | ✗ | ✗ | ✗ |
| WALLE | ✓ | ✓ | ✓ | ✓ | ✗ | ✗ | ✗ | ✗ |
| EpiScan | ✓ | ✗ | ✗ | ✗ | ✗ | ✗ | ✗ | ✗ |
| MIPE | ✓ | ✓ | ✗ | ✓ | ✓ | ✗ | ✗ | ✓ |
| **EpiFormer** | ✓ | ✓ | ✓ | ✓ | ✓ | ✓ | ✓ | ✓ |

**Discussion:** *EpiFormer* combines relation-aware EGNN-R message passing with early cross-attention to capture local structural detail and long-range inter-chain interactions. EGNN-R maintains E(3)-equivariance while encoding multi-relational protein structures, enabling invariant, geometry-aware representations under rigid-body transformations. Cross-attention in the encoder supports dynamic information exchange between antigen and antibody, providing binding context unavailable to antibody-agnostic approaches. The architecture jointly models intra-chain geometry and inter-chain binding dynamics using parallel processing streams, addressing a common limitation of prior methods. Together, these components allow simultaneous reasoning over structure and interaction without sacrificing equivariance. Performance on the epitope-group split suggests improved generalization to unseen binding sites, indicating that the model captures principles of antibody–antigen recognition rather than memorizing specific patterns. The method also produces interpretable antigen–antibody interaction maps: by modeling the full contact interface rather than only epitope residues, it predicts how binding is distributed across the paratope–epitope interface and highlights potential interaction hotspots.

**Limitations:** Despite the promising results by *EpiFormer*, there remain various ways to improve our model. Though we employ an E(3)-equivariant GNN in the encoder, exploring other alternatives such as *SE(3)*-equivariant GNNs (Fuchs et al., 2020) could improve its ability to handle global and local 3D symmetries. Our model can also be extended by performing self-supervised warm-up and transfer learning from general protein complexes that could boost its generalization capability (Zhang et al., 2022).

## 4.2 ABLATIONS

We conducted extensive ablations to isolate the contribution of each model component (please refer to Appendix A.5 for further details). Our analysis demonstrated that multi-relational graph structures substantially exceed the performance of basic proximity graphs (Table 4). Among the tested PLMs, ESM2-650M achieved the best results, outperforming both smaller and larger parameter variants (Table 5). The cross-attention-based decoders achieve 7.5% higher AUC than simple dot-product alternatives and maintain a better precision-recall balance (Table 6). The top-2 pooling strategy achieved superior performance over hierarchical (0.836), max (0.830), mean (0.834), and larger top-$k$ variants (Table 7). Our joint loss formulation which includes edge reconstruction, node classification, and auxiliary distance supervision achieves the best overall performance, while the failure of contrastive learning illustrates the challenges of multi-objective optimization in node classification tasks (Table 8). The most effective architectural configuration consisted of EGNN-R encoders paired with cross-attention decoders and the top-2 pooling strategy.

**GNNs:** To assess the impact of geometric message passing on epitope prediction performance, we systematically replaced the EGNN-R layers in the encoder of *EpiFormer* with alternative GNN architectures. We evaluated standard GNN variants including graph convolutional network (GCN) (Kipf, 2016), graph isomorphism network (GIN) (Xu et al., 2018), graph attention transformer (GAT) (Veličković et al., 2017), as well as more sophisticated approaches such as relational graph convolutional network (RGCN) (Zhang et al., 2022), and relation-aware equivariant graph network (REGNN) (Wu et al., 2025). As shown in Table 3, EGNN-R achieves superior performance across all metrics, with particularly notable improvements in AUPRC (0.443 vs 0.334 for REGNN) and F1 score (0.433 vs 0.343 for REGNN). While traditional GNNs like GCN, GIN, and GAT perform competitively but below EGNN-R, which highlights the critical importance of incorporating geometric equivariance for accurate modeling of three-dimensional protein binding interfaces.

Table 3: Performance comparison of different GNNs used in the *EpiFormer* encoder blocks on epitope prediction tasks. The best values are represented in bold, while the second-best values are underlined.

| Model | AUC | AUPRC | F1 | MCC | Precision | Recall |
|---|---|---|---|---|---|---|
| EGNN-R | **0.889 ± 0.045** | **0.443 ± 0.130** | **0.433 ± 0.014** | **0.404 ± 0.235** | **0.329 ± 0.067** | **0.633 ± 0.030** |
| GAT | 0.827 ± 0.006 | 0.308 ± 0.021 | 0.326 ± 0.010 | 0.276 ± 0.012 | 0.263 ± 0.016 | 0.435 ± 0.062 |
| GCN | 0.831 ± 0.006 | 0.325 ± 0.009 | 0.337 ± 0.010 | 0.290 ± 0.010 | 0.264 ± 0.014 | 0.467 ± 0.016 |
| GIN | 0.826 ± 0.007 | 0.310 ± 0.022 | 0.333 ± 0.016 | 0.284 ± 0.019 | 0.270 ± 0.004 | 0.437 ± 0.043 |
| REGNN | 0.833 ± 0.005 | 0.334 ± 0.015 | 0.343 ± 0.015 | 0.294 ± 0.015 | 0.276 ± 0.025 | 0.453 ± 0.016 |
| RGCN | 0.824 ± 0.004 | 0.314 ± 0.016 | 0.325 ± 0.008 | 0.276 ± 0.009 | 0.255 ± 0.018 | 0.452 ± 0.042 |

## 5 CONCLUSION

We presented *EpiFormer*, an encoder–decoder architecture for antibody-aware epitope prediction. Under comparable experimental conditions, *EpiFormer* outperforms prior methods on the AsEP benchmark and on the epitope-group split. Our experiments suggest that coupling multi-relational geometric message passing with cross-attention at different levels is a promising direction for antibody-specific epitope prediction. Extensive ablations demonstrate the robustness of our work.

## REPRODUCIBILITY STATEMENT

We will make the code publicly available on GitHub and provide installation scripts to address libraries' complex dependency issue. We hope that this will support and accelerate future research and development.

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

# A APPENDIX

## A.1 E(3)-EQUIVARIANCE OF THE EGNN-R LAYER

**Theorem 1** (E(3)-equivariance of the EGNN-R layer). *Consider the EGNN-R layer in §3.2 with updates*

$$m_{ij}^\rho = \phi_m^\rho(\mathbf{h}_i^\ell, \mathbf{h}_j^\ell, \gamma(d_{ij}), \mathbf{f}_{ij}), \tag{20}$$

$$s_{ij}^\rho = \phi_x^\rho(m_{ij}^\rho), \tag{21}$$

$$\mathbf{h}_i^{(\ell+1)} = \mathbf{h}_i^\ell + \phi_h\Big(\mathbf{h}_i^\ell, \sum_{j \in \mathcal{N}(i)} \sum_{\rho \in \mathbf{r}_{ij}} m_{ij}^\rho\Big), \tag{22}$$

$$\mathbf{x}_i^{(\ell+1)} = \mathbf{x}_i^\ell + \sum_{j \in \mathcal{N}(i)} \sum_{\rho \in \mathbf{r}_{ij}} \frac{\boldsymbol{\delta}_{ij}}{\sqrt{d_{ij} + \varepsilon}} \, s_{ij}^\rho, \tag{23}$$

*where $\boldsymbol{\delta}_{ij} = \mathbf{x}_i^\ell - \mathbf{x}_j^\ell$, $d_{ij} = \|\boldsymbol{\delta}_{ij}\|_2^2$, and $\varepsilon > 0$. Assume: (i) node features $\mathbf{h}_i^\ell \in \mathbb{R}^{d_h}$ are scalar channels, (ii) $\mathbf{h}_{ij}$ and $\mathbf{r}_{ij}$ are categorical and independent of coordinates, (iii) $\gamma$ is any scalar function of $d_{ij}$, (iv) each $\phi_{\{m,x\}}^\rho$ is an MLP from scalars to scalars. Let the $E(3)$ action be $g = (R, t)$ with $R \in O(3)$ and $t \in \mathbb{R}^3$, acting as $\mathbf{x}_i^\ell \mapsto R\mathbf{x}_i^\ell + t$ and $\mathbf{h}_i^\ell \mapsto \mathbf{h}_i^\ell$. Then the layer is $E(3)$-equivariant:*

$$\{\, \mathbf{x}_i^\ell, \mathbf{h}_i^\ell \,\}_{i=1}^n \mapsto \{\, R\mathbf{x}_i^\ell + t,\, \mathbf{h}_i^\ell \,\}_{i=1}^n \implies \{\, \mathbf{x}_i^{(\ell+1)}, \mathbf{h}_i^{(\ell+1)} \,\}_{i=1}^n \mapsto \{\, R\mathbf{x}_i^{(\ell+1)} + t,\, \mathbf{h}_i^{(\ell+1)} \,\}_{i=1}^n.$$

*Consequently, any stack of such layers is $E(3)$-equivariant by composition.*

*Proof.* Let $g = (R, t) \in E(3)$ act as stated. Edge data $\mathbf{f}_{ij}$ and $\mathbf{r}_{ij}$ are unchanged.

*Invariants.* Relative displacement and distance transform as

$$\boldsymbol{\delta}_{ij} \mapsto R\boldsymbol{\delta}_{ij}, \qquad d_{ij} = \|\boldsymbol{\delta}_{ij}\|^2 \mapsto \|R\boldsymbol{\delta}_{ij}\|^2 = d_{ij}. \tag{24}$$

Hence $d_{ij}$, $\gamma(d_{ij})$, and $(d_{ij} + \varepsilon)^{-1/2}$ are invariant scalars.

*Scalar messages and coefficients.* Each message $m_{ij}^\rho = \phi_m^\rho(\mathbf{h}_i^\ell, \mathbf{h}_j^\ell, \gamma(d_{ij}), \mathbf{f}_{ij})$ depends only on scalars that are invariant under $g$, so $m_{ij}^\rho$ is invariant. Then $s_{ij}^\rho = \phi_x^\rho(m_{ij}^\rho)$ is also invariant.

*Feature update.* The update

$$\mathbf{h}_i^{(\ell+1)} = \mathbf{h}_i^\ell + \phi_h\Big(\mathbf{h}_i^\ell, \sum_{j \in \mathcal{N}(i)} \sum_{\rho \in \mathbf{r}_{ij}} m_{ij}^\rho\Big) \tag{25}$$

uses only invariant scalars, so $\mathbf{h}_i^{(\ell+1)}$ is invariant. This matches the scalar action on features.

*Coordinate update.* The increment

$$\Delta\mathbf{x}_i = \sum_{j \in \mathcal{N}(i)} \sum_{\rho \in \mathbf{r}_{ij}} \frac{\boldsymbol{\delta}_{ij}}{\sqrt{d_{ij} + \varepsilon}} \, s_{ij}^\rho \tag{26}$$

is a sum of relative vectors scaled by invariant scalars. Under $g$ each term becomes

$$\frac{\boldsymbol{\delta}_{ij}}{\sqrt{d_{ij} + \varepsilon}} \, s_{ij}^\rho \mapsto \frac{R\boldsymbol{\delta}_{ij}}{\sqrt{d_{ij} + \varepsilon}} \, s_{ij}^\rho = R\Big(\frac{\boldsymbol{\delta}_{ij}}{\sqrt{d_{ij} + \varepsilon}} \, s_{ij}^\rho\Big), \tag{27}$$

so $\Delta\mathbf{x}_i \mapsto R\,\Delta\mathbf{x}_i$. Therefore

$$\mathbf{x}_i^{(\ell+1)} = \mathbf{x}_i^\ell + \Delta\mathbf{x}_i \mapsto R\mathbf{x}_i^\ell + t + R\Delta\mathbf{x}_i = R(\mathbf{x}_i^\ell + \Delta\mathbf{x}_i) + t = R\mathbf{x}_i^{(\ell+1)} + t. \tag{28}$$

*Composition.* The composition of equivariant maps is equivariant. Hence any stack of EGNN-R layers is $E(3)$-equivariant. $\square$

## A.2 MULTI-HEAD CROSS-ATTENTION WITH FEED-FORWARD NETWORK (MHCA)

The bidirectional multi-head cross-attention mechanism enables information exchange between antigen and antibody chains. Let $n_{\text{head}}$ be the number of heads with per-head width $d_a = d_h/n_{\text{head}}$. For layer $\ell$, independent linear projections produce queries, keys, and values:

$$\mathbf{Q}_{\text{ag}}^\ell = \mathbf{H}_{\text{ag}}^{(\ell-1)}\mathbf{W}_{\text{ag}}^{Q(\ell)}, \tag{29}$$

$$\mathbf{K}_{\text{ab}}^\ell = \mathbf{H}_{\text{ab}}^{(\ell-1)}\mathbf{W}_{\text{ab}}^{K(\ell)}, \tag{30}$$

$$\mathbf{V}_{\text{ab}}^\ell = \mathbf{H}_{\text{ab}}^{(\ell-1)}\mathbf{W}_{\text{ab}}^{V(\ell)}, \tag{31}$$

with analogous expressions for the reverse direction. After reshaping to $n_{\text{head}}$ heads of width $d_h$, scaled dot-product attention computes the affinity matrices:

$$\mathbf{A}_{\text{ag}\leftarrow\text{ab}}^\ell = \text{softmax}\Big(\frac{1}{\sqrt{d_h}}\,\mathbf{Q}_{\text{ag}}^\ell{\mathbf{K}_{\text{ab}}^\ell}^\top + \mathbf{M}\Big), \tag{32}$$

where $\mathbf{M}$ is a batch mask (applied only in decoder) that assigns $-\infty$ to residue pairs from different complexes. The resulting context vectors are:

$$\widetilde{\mathbf{H}}_{\text{ag}}^\ell = [\mathbf{A}_{\text{ag}\leftarrow\text{ab}}^\ell \mathbf{V}_{\text{ab}}^\ell]\mathbf{W}_{O,\text{ag}}^\ell, \tag{33}$$

$$\widetilde{\mathbf{H}}_{\text{ab}}^\ell = [\mathbf{A}_{\text{ab}\leftarrow\text{ag}}^\ell \mathbf{V}_{\text{ag}}^\ell]\mathbf{W}_{O,\text{ab}}^\ell. \tag{34}$$

Each direction then applies a feed-forward network $\text{FFN}(\mathbf{x}) = \mathbf{W}_2\,\sigma(\mathbf{W}_1\mathbf{x}+\mathbf{b}_1)+\mathbf{b}_2$ with dropout, residual connections, and layer normalization.

## A.3 GRAPH CONSTRUCTION

### A.3.1 NODE FEATURES

Each residue node in our protein graph incorporates two complementary information sources that together provide a rich representation of both local structural properties and evolutionary context:

**Local geometry & physicochemistry:** Each residue $v_i \in \mathcal{V}$ is annotated with a 105-dimensional geometric and biochemical feature vector $\mathbf{h}_i^{\text{geo}} \in \mathbb{R}^{d_{\text{geo}}}$ that encodes the type, position, distance, direction, angle, and orientation of each residue. Such residue-level descriptors are widely employed in diverse protein-related studies in structural bioinformatics (Wu et al., 2025; Jing et al., 2020; Jumper et al., 2021). This vector is constructed as follows:

$$\mathbf{h}_i^{\text{geo}} = \Big[E_{\text{type}}(v_i),\ E_{\text{pos}}(i),\ \sin(\eta_i),\ \cos(\eta_i),\ \text{RBF}(\|\mathbf{x}_{i,C_\alpha} - \mathbf{x}_{i,\xi}\|),\ Q_i^\top \frac{\mathbf{x}_{i,\xi} - \mathbf{x}_{i,C_\alpha}}{\|\mathbf{x}_{i,\xi} - \mathbf{x}_{i,C_\alpha}\|}\Big], \tag{35}$$

where:

- $E_{\text{type}}$: Embedding for amino acid residue type (e.g., arginine, glycine).

- $E_{\text{pos}}$: Positional encoding of residue index in the sequence, enabling the model to distinguish between identical amino acids based on their sequence context. This positional information is crucial for understanding long-range dependencies and structural motifs, as amino acids at different sequence positions (N-terminus vs. C-terminus, loop regions vs. secondary structures) often play different functional roles even if they are the same amino acid type.

- $\eta_i$: Local backbone geometry encoded through six fundamental angles that determine how the protein chain folds at each residue $v_i$ and are encoded by their sine and cosine (12 scalars). Bond angles $(\alpha_i, \beta_i, \gamma_i)$ describe the geometric constraints of covalent bonds, while dihedral angles $(\psi_i, \phi_i, \omega_i)$ capture the rotational freedom that gives rise to secondary structures like helices and sheets.

- $\text{RBF}(\cdot)$: Radial basis function encoding distances between $C_\alpha$ and other backbone atoms ($\xi \in \{C_\beta, N, O\}$), with each distance represented by 16 Gaussian basis functions.

- $Q_i^\top \mathbf{u}_i$: Here, $Q_i \in \mathbb{R}^{3\times 3}$ is the orthonormal rotation matrix defining the local coordinate system constructed from the $C_\alpha$, $C_\beta$, and N atoms of residue $i$, and $\mathbf{u}_i = [\mathbf{u}_i^1, \mathbf{u}_i^2, \mathbf{u}_i^3] \in \mathbb{R}^{3\times 3}$ contains the normalized direction vectors between these atoms (e.g., $\mathbf{u}_i^1 = \frac{\mathbf{x}_{i,C_\beta} - \mathbf{x}_{i,C_\alpha}}{\|\mathbf{x}_{i,C_\beta} - \mathbf{x}_{i,C_\alpha}\|}$). The matrix product $Q_i^\top \mathbf{u}_i$ transforms these direction vectors into the local coordinate frame and is flattened to yield a 9-dimensional feature vector. Note that the oxygen atom is stored in the coordinate matrix for other calculations (like the RBF distance features), but isn't used for the local coordinate frame construction.

- The coordinates are held in a $3 \times 4$ matrix which is used in the calculation of node and edge features.

$$\mathbf{X}_i = \begin{bmatrix} \mathbf{x}_{i,\text{N}} & \mathbf{x}_{i,C_\alpha} & \mathbf{x}_{i,C_\beta} & \mathbf{x}_{i,\text{O}} \end{bmatrix} \quad \in \mathbb{R}^{3\times 4}, \quad \text{where} \quad \mathbf{x}_{i,\xi} \in \mathbb{R}^3$$

**Frozen protein-language-model (PLM) embeddings** We extract embeddings for the antigen and antibody sequences $\mathbf{z}_i^{\text{plm}} \in \mathbb{R}^{d_c}$ using pre-trained protein-language models (e.g., ESM-2 (Lin et al., 2023)) to provide the model an orthogonal information source (evolutionary + biochemical context). Since the original PLM embeddings are high-dimensional (for example, $d_c = 1280$ for ESM2-650M), we project them to a lower-dimensional representation suitable for our architecture:

$$\mathbf{h}_i^{\text{plm}} = \mathbf{W}_{\text{plm}}\mathbf{z}_i^{\text{plm}}, \quad \text{where} \quad \mathbf{W}_{\text{plm}} \in \mathbb{R}^{d_{\text{plm}} \times d_c}. \tag{36}$$

Here, $d_{\text{plm}}$ is the target dimensionality for the compressed PLM features, and $\mathbf{W}_{\text{plm}}$ serves as a learnable bottleneck that adapts the frozen PLM representations to our specific task.

### A.3.2 EDGE FEATURES

We compute a 100-dimensional edge feature vector $\mathbf{f}_{i,j} \in \mathbb{R}^{d_h}$ that describes the spatial and sequential relationship between two residues $v_i$ and $v_j$. This vector integrates multiple complementary descriptors to provide a rich representation of inter-residue interactions (Jing et al., 2020) and is defined as follows:

$$\mathbf{f}_{i,j} = \left\{ E_{\text{type}}(e_{i,j}), E_{\text{pos}}(i-j), \text{RBF}(\|\mathbf{x}_{i,C_\alpha} - \mathbf{x}_{j,\xi}\|), Q_i^\top \frac{\mathbf{x}_{j,\xi} - \mathbf{x}_{i,C_\alpha}}{\|\mathbf{x}_{j,\xi} - \mathbf{x}_{i,C_\alpha}\|}, q\left(Q_i^\top Q_j\right) \mid \xi \right\}, \tag{37}$$

where $E_{\text{type}}(e_{i,j})$ is the one-hot encoding of relations $\mathbf{r}_{i,j}$ of length 4 between two residues, and the positional encoding $E_{\text{pos}}(i-j)$ encodes the relative sequential position sinusoidally to 16 scalars. The third and fourth terms are distance and direction encodings of four backbone atoms $\xi$ in residue $v_j$ in the local coordinate frame $Q_i$. These four inter-residue distances $\{d(C_\alpha, C_\beta), d(C_\alpha, \text{N}), d(C_\alpha, \text{O}), d(C_\alpha, C_\alpha)\}$) are each represented by 16 Gaussian basis functions. The last term $q\left(Q_i^\top Q_j\right)$ is the quaternion representation $q(\cdot)$ of $Q_i^\top Q_j$. By integrating sequence position, local geometry, and orientation, the model understands the residue identity from global pose and enables robust generalization across structures. These node and edge features are visualized in Figure 3(a).

### A.3.3 EDGE RELATIONS

Since spatial proximity between residues alone cannot capture hydrogen bonding's directional specificity or electrostatic complementarity's charge-based selectivity, we use multi-relational edges to capture distinct interaction types (Zhang et al., 2022). By treating each relation separately, the model learns complex interaction patterns within the protein. Hence, to expand the contexts of these interactions, we divide the edges into four different types of relations $\mathcal{R} = \{\rho_1, \rho_2, \rho_3, \rho_4\}$, including *(i)* **sequential relations** $\rho_1$ and $\rho_2$ between two residues with relative sequential distance equal to 1 (peptide bond) and 2 (short-range torsion coupling); *(ii)* **spatial relations** between residues that are from the same component and spatially connected due to $K$-nearest neighbors (relation $\rho_3$ that captures local packing shell) or with a Euclidean distance less than 8Å (relation $\rho_4$) capturing medium-range contact between residues within the protein structure (Wu et al., 2025).

To illustrate the importance of edge relations, consider a discontinuous epitope spanning two antigen loops: sequential edges ($\rho_1, \rho_2$) maintain the structural integrity of each loop, while spatial edges ($\rho_3, \rho_4$) capture the three-dimensional proximity between residues from different loops, enabling the model to understand how distant sequence regions come together to form a cohesive binding interface. We provide a schematic of edge relations in Fig. 3 (b), where each edge $e_{i,j} \in \mathcal{E}$ is associated with a

set of relations $\mathbf{r}_{i,j} \in \mathcal{R}$. Besides, two relations $\rho_1$ (with sequence distance equal to 1) can derive a relation $\rho_2$ (with sequence distance equal to 2), while an edge may connect two nodes (residues) due to both relations $\rho_3$ and $\rho_4$.

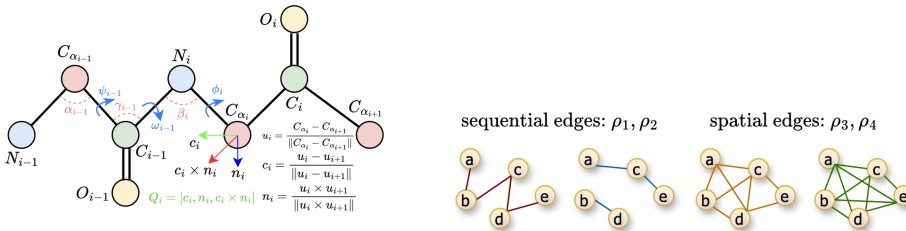

Figure 3: (a) Node and edge features encoding position, distance, direction, angle, and orientation (Figure credit: (Wu et al., 2025)). (b) Four edge relations (sequential $\rho_1$, $\rho_2$; spatial $\rho_3$, $\rho_4$ ). To avoid complexity, we visualize only some edges.

### A.3.4 Preprocessing

For each complex, we first separated the paired antigen and antibody chains into individual structure files. We then performed sequence-structure alignment using Clustal Omega (Sievers et al., 2011) to establish correspondence between SEQRES (complete sequence) and ATOMSEQ (resolved atoms) records. This alignment generated binary masks that enable reliable mapping of sequences to structural residues (seqres2surf and seqres2cdr) while preserving the native crystallographic ordering.

For antibody chains, we applied the alignment masks to reindex heavy (H) and light (L) chains by removing insertion codes to enforce consecutive 1-based residue numbering required for graph construction. Antigen chains underwent similar processing to maintain parity between sequences and structures. This step ensures that each residue in the protein sequence corresponds exactly to its structural counterpart during the graph representation. Then, we applied solvent-accessibility filters to retain only antigen surface residues, using the original AsEP seqres2surf masks to define the node set for antigen residue graphs. The binary epitope labels were projected onto the surface ATOMSEQ via alignment masks, while paratope labels were preserved for antibody residue nodes. This surface filtering step prevents non-surface residues from confounding epitope supervision while maintaining all necessary information for cross-chain interaction modeling.

To incorporate evolutionary and semantic information, we integrated embeddings from state-of-the-art PLMs. For antigens, we extracted embeddings using the ESM model family, while, for antibodies, we incorporated AntiBERTy embeddings (Ruffolo et al., 2023; Ahmed et al., 2025), a transformer model specialized for antibody sequences, providing better functional and evolutionary context for paratope regions. These embeddings were mapped to graph nodes using the seqres2atmseq alignment masks. Finally, we used these preprocessed structures to generate HeteroData objects for the multi-relational graphs using PyTorch Geometric (Fey & Lenssen, 2019).

### A.4 Implementation Details

The model is trained with an Adam optimizer and a ReduceLROnPlateau learning-rate schedule with decoupled weight decay. The learning rate is selected from the sweep-defined range and fixed at approximately $9.1e-5$ in the best configuration. A ReduceLROnPlateau scheduler monitors validation performance and decays the learning rate on stagnation, while an early stopping with patience of 10 epochs prevents overfitting and reduces variance in final selection. We used SiLU activation functions (Elfwing et al., 2018) throughout the model because they provide stable gradients via their smooth, non-monotonic curve, which are crucial for training deep graph networks. The hyperparameter tuning was performed via a Bayesian optimization sweep in Weights & Biases to maximize validation F1 score, and the best hyperparameters were chosen within a predefined search space using bounded uniform and log-uniform distributions.

- The model weight decay was sampled log-uniformly over $[1e-5, 1e-6]$ to prevent overfitting by penalizing large weights.

- The model dropout was sampled log-uniformly over $[0.05, 0.5]$ to improve the generalizability of the model, and the best performing configuration used a dropout of 0.132.

- The number of layers in the encoder module is treated as a hyperparameter and was chosen from the set $[3, 4, 5]$ while for the decoder, the number of layers was chosen from the $[2, 3, 4]$. We experimented with different encoder hidden dimensions and the best configuration of 128 was picked from $[64, 128, 256, 512]$ across different runs.

- We also experimented with different number of attention heads for the encoder and decoder MHCA (2,4,8,16) and picked the best model with 8 attention heads.

- A batch size of 8 was chosen from [4,8,16,32] across different runs.

- $\alpha_{\mathrm{ag}}$ and $\alpha_{\mathrm{ab}}$ are initialised to 0.05

For the loss coefficients, the best run uses $\lambda_{\mathrm{edge}} = 1.0$, $\lambda_{\mathrm{node}} = 0.4816$, $\lambda_{\mathrm{geo}} = 0.0514$, $\beta_{\mathrm{BCE}} = 9.3249$, $\beta_{\mathrm{Dice}} = 2.2966$, $\beta_{\mathrm{sparsity}} = 0.3068$, $\pi_{\mathrm{epi}} = 15.2856$, $\pi_{\mathrm{edge}} = 58.7077$, label smoothing $\epsilon = 0.1$, and a distance cutoff of 32 Å for $\mathcal{L}_{\mathrm{geo}}$.

- The bipartite edge positive-class weight $\pi_{\mathrm{edge}}$ for the BCE-with-logits interaction loss was sampled log-uniformly over $[30, 150]$, accommodating variation in pairwise sparsity across complexes.

- The node objective weight $\lambda_{\mathrm{node}}$ was sampled uniformly over $[0.05, 0.5]$, exploring the trade-off between residue supervision and the other objectives.

- The binary cross-entropy multiplier within the node objective $\beta_{\mathrm{BCE}}$ was drawn uniformly over $[2, 10]$, spanning weak to strong emphasis on classification error.

- The Dice multiplier $\beta_{\mathrm{Dice}}$ was drawn uniformly over $[0.1, 3.0]$, reflecting its role as a secondary calibrator under class imbalance.

- The epitope positive-class weight $\pi_{\mathrm{epi}}$ was sampled log-uniformly over $[10, 60]$, covering roughly an order of magnitude in imbalance without biasing toward either extreme.

- The per-graph epitope count-regularizer weight $\beta_{\mathrm{sparsity}}$ was sampled uniformly over $[0.05, 1.0]$, enabling calibration of predicted positive counts at the complex level.

- The auxiliary distance-classification weight $\lambda_{\mathrm{geo}}$ was sampled uniformly over $[0.05, 0.3]$, with class balancing across distance bins and distance-aware pair weighting kept enabled and the maximum distance fixed at 32 Å for all trials.

The experiments were performed on an NVIDIA RTX 6000 GPU and it took around 35-60 minutes for a single hyperparameter sweeping experiment of around 50 epochs. To ensure full reproducibility of our experiments, we implement random seed management across all computational components including NumPy (`numpy.random`), Python (`random`), PyTorch (`torch.manual_seed`), and CUDA operations (`torch.cuda.manual_seed_all`), while additionally controlling worker initialization in data loaders and disabling non-deterministic algorithms (`torch.backends.cudnn.deterministic=True`).

## A.5 ABLATION STUDIES

We performed ablation studies on the different protein graph representations, model components such as encoder and decoder architectures, pooling strategies, and loss functions. The results are reported as mean ± standard deviation over 3 random seeds.

### A.5.1 GRAPH CONSTRUCTION

This ablation isolates how residue-level graph design affects *EpiFormer*'s antibody-specific epitope prediction by holding node/edge features, PLM inputs, and training configuration fixed while swapping the underlying graph topology. Specifically, we compared three protein graph representations: a simple residue-only graph that collapses relations into proximity edges (Choi & Kim, 2024), a RAAD-style multi-relational graph with four edge types (sequential and spatial) (Wu et al., 2025), and a GearNet (Zhang et al., 2022) variant with seven relation types constructed to capture finer-grained structural neighborhoods. The node and edge features were fixed for all three graph types, and

the edge relations were only varied. This design quantifies the contribution of relation granularity and edge semantics of the proteins to the downstream performance of epitope prediction. Table 4 compares the epitope prediction performance of *EpiFormer* using the three graph representations.

Table 4: Performance metrics for different protein graph representation architectures on epitope prediction tasks. All values are reported for the epitope-to-surface ratio split. The best values are represented in bold, while the second-best values are underlined.

| Graph type | AUC | AUPRC | F1 | MCC | Precision | Recall |
|---|---|---|---|---|---|---|
| Simple | 0.821 | 0.355 | 0.333 | 0.294 | 0.240 | 0.543 |
| GearNet | 0.812 | 0.315 | 0.337 | 0.286 | 0.290 | 0.401 |
| Multi-relational | **0.888** | **0.443** | **0.433** | **0.404** | **0.329** | **0.633** |

We also performed experiments by using different sequence embeddings from the Evolutionary Scale Modeling (ESM) family to explore their contribution to the epitope prediction task. We used three variants of the ESM2 (Lin et al., 2023) model family (35M, 650M, and 3B parameters) as well as the newer ESM3-small (Hayes et al., 2025) model (1.4B parameters). Our experiments in Table 5 show that ESM2-650M produces the best contextual features for the antigen-antibody binding site prediction task.

Table 5: Performance metrics for different PLM embeddings on the epitope prediction tasks. 4 models from the Evolutionary Scale Modeling (ESM) family were used to generate embeddings for antigens, while AntiBERTy (IgFold) was used to generate embeddings for the antibodies. All values are reported for the epitope-to-surface ratio split. The best values are represented in bold, while the second-best values are underlined.

| PLM | AUC | AUPRC | F1 | MCC | Precision | Recall |
|---|---|---|---|---|---|---|
| ESM2-35M | 0.815 | 0.330 | 0.334 | 0.283 | 0.287 | 0.399 |
| ESM2-650M | **0.888** | **0.443** | **0.433** | **0.404** | **0.329** | **0.633** |
| ESM2-3B | 0.826 | 0.331 | 0.349 | 0.300 | 0.285 | 0.449 |
| ESM3-small | 0.840 | 0.374 | 0.377 | 0.330 | 0.331 | 0.437 |

### A.5.2 MODEL

We also replaced the cross-attention decoder with dot-product and dual alternatives. The **dot-product** decoder computes the interaction matrix as a plain inner product between antigen and antibody embeddings and produces a fast and parameter-free similarity score. The **dual** decoder architecture integrates two parallel processing paths: a dot-product similarity route and a sparse cross-attention mechanism, and merges their outputs via a learnable weight $\alpha$. The ablation studies show lower AUC, AUPRC, and F1 metrics for dot product decoders compared to cross-attention and dual decoders. Dot-product decoding favors precision but substantially reduces recall, whereas cross-attention preserves a stronger precision–recall balance as shown in Table 6.

Table 6: Performance comparison of different decoder blocks for epitope prediction. The best values are represented in bold, while the second-best values are underlined.

| Decoder | AUC | AUPRC | F1 | MCC | Precision | Recall |
|---|---|---|---|---|---|---|
| Cross Attn. | **0.889 $\pm$ 0.045** | **0.443 $\pm$ 0.130** | **0.433 $\pm$ 0.014** | **0.404 $\pm$ 0.235** | **0.329 $\pm$ 0.067** | **0.633 $\pm$ 0.030** |
| Dot Product | 0.827 $\pm$ 0.009 | 0.315 $\pm$ 0.034 | 0.326 $\pm$ 0.011 | 0.278 $\pm$ 0.015 | 0.252 $\pm$ 0.009 | 0.464 $\pm$ 0.053 |
| Dual | 0.834 $\pm$ 0.008 | 0.339 $\pm$ 0.030 | 0.334 $\pm$ 0.014 | 0.286 $\pm$ 0.017 | 0.266 $\pm$ 0.008 | 0.450 $\pm$ 0.033 |

We performed ablation studies over different pooling strategies. We map the bipartite interaction matrix $\hat{\mathcal{E}}_{bg}$ to per-residue probabilities by aggregating across the partner dimension (row-wise for epitopes, column-wise for paratopes): **Max pooling** assigns the maximum interaction per residue; **Mean pooling** averages interactions over all partners; **Top-$k$ mean pooling** averages the largest $k$ interactions (small $k$, e.g., 2) to reflect a few key partners; **Noisy-OR** aggregates as $1 - \prod_j (1 - Y_{ij})$, modeling the probability that at least one partner induces a positive signal; **Softmax-attention**

converts interactions to attention weights via a softmax along the partner dimension and returns the weighted sum; **Hierarchical pooling** takes a convex combination of top-2 mean (local specificity) and global mean (context) with a mixing weight $\alpha$. Empirically (Table 7), Top-2 pooling yields the highest AUC/AUPRC/F1, hierarchical pooling is competitive, while max/mean/softmax-attention and larger $k$ underperform and tend to over-concentrate probability mass and impair calibration.

Table 7: Performance comparison of different pooling methods for epitope prediction. The best values are represented in bold, while the second-best values are underlined.

| Pooling Method | AUC | AUPRC | F1 | MCC | Precision | Recall |
|---|---|---|---|---|---|---|
| Hierarchical Pooling | $0.836 \pm 0.004$ | $0.338 \pm 0.012$ | $0.341 \pm 0.009$ | $0.295 \pm 0.006$ | $0.268 \pm 0.022$ | $0.476 \pm 0.038$ |
| Max | $0.830 \pm 0.005$ | $0.321 \pm 0.021$ | $0.326 \pm 0.006$ | $0.279 \pm 0.011$ | $0.265 \pm 0.029$ | $0.441 \pm 0.085$ |
| Mean | $0.834 \pm 0.006$ | $0.324 \pm 0.016$ | $0.332 \pm 0.004$ | $0.283 \pm 0.004$ | $0.281 \pm 0.024$ | $0.414 \pm 0.048$ |
| Pool Top-2 | $\mathbf{0.889 \pm 0.045}$ | $\mathbf{0.443 \pm 0.130}$ | $\mathbf{0.433 \pm 0.014}$ | $\mathbf{0.404 \pm 0.235}$ | $\mathbf{0.329 \pm 0.067}$ | $\mathbf{0.633 \pm 0.030}$ |
| Pool Top-3 | $0.851 \pm 0.030$ | $0.370 \pm 0.062$ | $0.370 \pm 0.048$ | $0.330 \pm 0.059$ | $0.286 \pm 0.034$ | $0.529 \pm 0.103$ |
| Pool Top-4 | $0.836 \pm 0.008$ | $0.342 \pm 0.019$ | $0.340 \pm 0.018$ | $0.295 \pm 0.019$ | $0.260 \pm 0.020$ | $0.493 \pm 0.020$ |
| Softmax Attn. | $0.832 \pm 0.007$ | $0.329 \pm 0.018$ | $0.332 \pm 0.004$ | $0.285 \pm 0.005$ | $0.256 \pm 0.006$ | $0.472 \pm 0.020$ |

### A.5.3 LOSS

We performed ablations to evaluate the contribution of the loss function/s (primary, auxiliary, and regularizers) on the epitope prediction task, as shown in Table 8.

**Contrastive Learning Loss ($\mathcal{L}_{\mathbf{InfoNCE}}$)** We also performed contrastive learning with the SimCLR InfoNCE (Information Noise Contrastive Estimation) loss (Chen et al., 2020) to learn discriminative representations by contrasting positive and negative residue pairs within and across protein chains. The contrastive loss combines intra-chain and inter-chain objectives:

$$\mathcal{L}_{\text{contrastive}} = \lambda_{\text{intra}}\mathcal{L}_{\text{intra}} + \lambda_{\text{inter}}\mathcal{L}_{\text{inter}}, \tag{38}$$

where $\lambda_{\text{intra}}$ and $\lambda_{\text{inter}}$ balance the relative importance of within-chain and cross-chain contrastive learning.

INTRA-CHAIN CONTRASTIVE LOSS ($\mathcal{L}_{\text{INTRA}}$) The intra-chain loss encourages similar representations for residues with the same label (epitope/non-epitope or paratope/non-paratope) within each protein chain:

$$\mathcal{L}_{\text{intra}} = \mathcal{L}_{\text{intra}}^{\text{ag}} + \mathcal{L}_{\text{intra}}^{\text{ab}}. \tag{39}$$

For each chain (antigen or antibody), the loss is computed as:

$$\mathcal{L}_{\text{intra}}^{\text{chain}} = -\frac{1}{|\mathcal{P}|} \sum_{i \in \mathcal{P}} \log \frac{\sum_{j \in \mathcal{P}_{i+}} \exp(\mathbf{h}_i^T \mathbf{h}_j / \tau)}{\sum_{k \in \mathcal{N}_i} \exp(\mathbf{h}_i^T \mathbf{h}_k / \tau)}, \tag{40}$$

where $\mathcal{P} = \{i : y_i = 1\}$ is the set of positive (binding) residues, $\mathcal{P}_{i+} = \{j \in \mathcal{P} : j \neq i\}$ are other positive residues sharing the same label as anchor $i$, $\mathcal{N}_i$ includes all negative residues for anchor $i$, $\mathbf{h}_i, \mathbf{h}_j$ are $L_2$-normalized residue embeddings, and $\tau$ is the temperature parameter controlling concentration.

INTER-CHAIN CONTRASTIVE LOSS ($\mathcal{L}_{\text{INTER}}$) The inter-chain loss promotes alignment between epitope and paratope representations across antigen-antibody pairs:

$$\mathcal{L}_{\text{inter}} = \mathcal{L}_{\text{ag} \rightarrow \text{ab}} + \mathcal{L}_{\text{ab} \rightarrow \text{ag}}. \tag{41}$$

The bidirectional formulation ensures symmetric learning:

$$\mathcal{L}_{\text{ag} \rightarrow \text{ab}} = -\frac{1}{|\mathcal{P}_{\text{ag}}|} \sum_{i \in \mathcal{P}_{\text{ag}}} \log \frac{\sum_{j \in \mathcal{P}_{\text{ab}}} \exp(\mathbf{h}_i^{\text{ag}T} \mathbf{h}_j^{\text{ab}} / \tau)}{\sum_{k \in \mathcal{N}_{\text{cross}}} \exp(\mathbf{h}_i^{\text{ag}T} \mathbf{h}_k / \tau)}, \tag{42}$$

where $\mathcal{P}_{\text{ag}}, \mathcal{P}_{\text{ab}}$ are epitope and paratope residue sets, $\mathcal{N}_{\text{cross}}$ includes negative residues from both chains, and the loss pulls epitope embeddings closer to paratope embeddings while pushing them away from non-binding residues. Our experiments show that contrastive learning didn't contribute

to improving the classification performance. We attribute this to conflicting optimization objectives between BCE loss and standard InfoNCE loss, a phenomenon demonstrated in a recent work (Ji et al., 2024).

Table 8: Performance comparison of different loss function configurations for epitope prediction. All metrics are reported for epitope prediction tasks. The best values are represented in bold, while the second-best values are underlined.

| Loss Configuration | AUC | AUPRC | F1 | MCC | Precision | Recall |
|---|---|---|---|---|---|---|
| $\mathcal{L}_{bce}$ | $0.822 \pm 0.013$ | $0.274 \pm 0.038$ | $0.199 \pm 0.017$ | $0.205 \pm 0.022$ | $0.111 \pm 0.011$ | $0.946 \pm 0.016$ |
| $\mathcal{L}_{edge}$ | $0.581 \pm 0.006$ | $0.098 \pm 0.002$ | $0.142 \pm 0.010$ | $0.086 \pm 0.007$ | $0.154 \pm 0.002$ | $0.132 \pm 0.016$ |
| $\mathcal{L}_{bce} + \mathcal{L}_{geo}$ | $0.822 \pm 0.009$ | $0.266 \pm 0.025$ | $0.205 \pm 0.012$ | $0.214 \pm 0.013$ | $0.115 \pm 0.008$ | $0.941 \pm 0.019$ |
| $\mathcal{L}_{bce} + \mathcal{L}_{edge}$ | $0.826 \pm 0.006$ | $0.296 \pm 0.018$ | $0.220 \pm 0.008$ | $0.230 \pm 0.009$ | $0.125 \pm 0.005$ | $0.914 \pm 0.011$ |
| $\mathcal{L}_{bce} + \mathcal{L}_{edge} + \mathcal{L}_{dice}$ | $0.818 \pm 0.011$ | $0.268 \pm 0.029$ | $0.205 \pm 0.017$ | $0.210 \pm 0.021$ | $0.115 \pm 0.011$ | $0.930 \pm 0.026$ |
| $\mathcal{L}_{bce} + \mathcal{L}_{edge} + \mathcal{L}_{geo}$ | $0.826 \pm 0.015$ | $0.299 \pm 0.050$ | $0.214 \pm 0.020$ | $0.223 \pm 0.022$ | $0.121 \pm 0.013$ | $0.926 \pm 0.033$ |
| $\mathcal{L}_{edge} + \mathcal{L}_{node} + \mathcal{L}_{geo}$ | $\mathbf{0.889 \pm 0.045}$ | $\mathbf{0.443 \pm 0.130}$ | $\mathbf{0.433 \pm 0.014}$ | $\mathbf{0.404 \pm 0.235}$ | $\mathbf{0.329 \pm 0.067}$ | $\mathbf{0.633 \pm 0.030}$ |
| $\mathcal{L}_{edge} + \mathcal{L}_{node} + \mathcal{L}_{geo} + \mathcal{L}_{InfoNCE}$ | $\underline{0.850 \pm 0.031}$ | $\underline{0.362 \pm 0.064}$ | $\underline{0.361 \pm 0.051}$ | $\underline{0.323 \pm 0.064}$ | $\underline{0.270 \pm 0.034}$ | $\underline{0.550 \pm 0.111}$ |
| $\mathcal{L}_{bce} + \mathcal{L}_{edge} + \mathcal{L}_{InfoNCE}$ | $0.837 \pm 0.002$ | $0.345 \pm 0.007$ | $0.338 \pm 0.008$ | $0.296 \pm 0.005$ | $0.254 \pm 0.020$ | $0.511 \pm 0.049$ |
| $\mathcal{L}_{bce} + \mathcal{L}_{edge} + \mathcal{L}_{sparsity}$ | $0.835 \pm 0.002$ | $0.336 \pm 0.013$ | $0.334 \pm 0.006$ | $0.288 \pm 0.007$ | $0.270 \pm 0.026$ | $0.453 \pm 0.078$ |
| $\mathcal{L}_{edge} + \mathcal{L}_{node}$ | $0.835 \pm 0.003$ | $0.325 \pm 0.012$ | $0.329 \pm 0.001$ | $0.283 \pm 0.004$ | $0.260 \pm 0.026$ | $0.462 \pm 0.071$ |
| $\mathcal{L}_{edge} + \mathcal{L}_{node} + \mathcal{L}_{InfoNCE}$ | $0.829 \pm 0.006$ | $0.305 \pm 0.015$ | $0.326 \pm 0.010$ | $0.276 \pm 0.012$ | $0.261 \pm 0.006$ | $0.435 \pm 0.023$ |

## LLM USAGE CLAIM

LLMs were used in a limited capacity for the retrieval and discovery of related work. During paper writing, LLMs were used for the purpose of improving grammar and wording. All technical content, experimental design, implementation, analysis, and scientific contributions are entirely the authors' original work.

---

**Algorithm 1:** EpiFormer: High-Level Architecture

---

**Input:** Antigen graph $\mathcal{G}_{\mathrm{ag}}$ and antibody graph $\mathcal{G}_{\mathrm{ab}}$ with coordinates $\mathbf{X}$, features $\mathbf{h}^{\mathrm{geo}}$, $\mathbf{h}^{\mathrm{plm}}$

**Output:** Bipartite interaction matrix $\hat{\mathcal{E}}_{\mathrm{bg}} \in [0,1]^{n \times m}$

    // **Feature Initialization**

1 **foreach** *chain* $\in \{ag, ab\}$ **do**

2     Apply gating network to combine geometric and PLM features;

3     $\mathbf{h}_i^0 \leftarrow \mathrm{Gate}(\mathbf{h}_i^{\mathrm{geo}}, \mathbf{h}_i^{\mathrm{plm}})$ for each residue $i$;

4 **end**

    // **Encoder: Parallel Processing**

5 **for** *layer* $\ell = 1$ **to** $L$ **do**

      // Intra-chain geometric message passing

6     $(\mathbf{H}_{\mathrm{ag}}^{\mathrm{intra}}, \mathbf{X}^{\mathrm{ag}}) \leftarrow \mathrm{EGNN\text{-}R}(\mathcal{G}_{\mathrm{ag}}, \mathbf{H}_{\mathrm{ag}}^{(\ell-1)}, \mathbf{X}^{\mathrm{ag}})$;

7     $(\mathbf{H}_{\mathrm{ab}}^{\mathrm{intra}}, \mathbf{X}^{\mathrm{ab}}) \leftarrow \mathrm{EGNN\text{-}R}(\mathcal{G}_{\mathrm{ab}}, \mathbf{H}_{\mathrm{ab}}^{(\ell-1)}, \mathbf{X}^{\mathrm{ab}})$;

      // Inter-chain cross-attention

8     $\widetilde{\mathbf{H}}_{\mathrm{ag}} \leftarrow \mathrm{MHCA}(\mathbf{H}_{\mathrm{ag}}^{\mathrm{intra}}, \mathbf{H}_{\mathrm{ab}}^{\mathrm{intra}}, \mathbf{H}_{\mathrm{ab}}^{\mathrm{intra}})$;

9     $\widetilde{\mathbf{H}}_{\mathrm{ab}} \leftarrow \mathrm{MHCA}(\mathbf{H}_{\mathrm{ab}}^{\mathrm{intra}}, \mathbf{H}_{\mathrm{ag}}^{\mathrm{intra}}, \mathbf{H}_{\mathrm{ag}}^{\mathrm{intra}})$;

      // Combine intra-chain and cross-chain information

10    $\mathbf{H}_{\mathrm{ag}}^{\ell} \leftarrow \mathbf{H}_{\mathrm{ag}}^{(\ell-1)} + \mathbf{H}_{\mathrm{ag}}^{\mathrm{intra}} + \alpha_{\mathrm{ag}} \, \mathrm{FFN}(\widetilde{\mathbf{H}}_{\mathrm{ag}})$;

11    $\mathbf{H}_{\mathrm{ab}}^{\ell} \leftarrow \mathbf{H}_{\mathrm{ab}}^{(\ell-1)} + \mathbf{H}_{\mathrm{ab}}^{\mathrm{intra}} + \alpha_{\mathrm{ab}} \, \mathrm{FFN}(\widetilde{\mathbf{H}}_{\mathrm{ab}})$;

12 **end**

    // **Decoder: Cross-Attention Refinement**

13 Initialize decoder embeddings: $\mathbf{H}_{\mathrm{ag}}^{\mathrm{dec}} \leftarrow \mathbf{H}_{\mathrm{ag}}^{L}$, $\mathbf{H}_{\mathrm{ab}}^{\mathrm{dec}} \leftarrow \mathbf{H}_{\mathrm{ab}}^{L}$;

14 **for** *layer* $\ell = 1$ **to** $L$ **do**

      // Inter-chain cross-attention

15    $\widetilde{\mathbf{H}}_{\mathrm{ag}}^{\mathrm{dec}} \leftarrow \mathrm{MHCA}(\mathbf{H}_{\mathrm{ag}}^{\mathrm{dec}}, \mathbf{H}_{\mathrm{ab}}^{\mathrm{dec}}, \mathbf{H}_{\mathrm{ab}}^{\mathrm{dec}})$;

16    $\widetilde{\mathbf{H}}_{\mathrm{ab}}^{\mathrm{dec}} \leftarrow \mathrm{MHCA}(\mathbf{H}_{\mathrm{ab}}^{\mathrm{dec}}, \mathbf{H}_{\mathrm{ag}}^{\mathrm{dec}}, \mathbf{H}_{\mathrm{ag}}^{\mathrm{dec}})$;

      // Combine intra-chain and cross-chain information

17    $\mathbf{H}_{\mathrm{ag}}^{\mathrm{dec}(\ell)} \leftarrow \mathbf{H}_{\mathrm{ag}}^{\mathrm{dec}(\ell-1)} + \mathrm{FFN}(\widetilde{\mathbf{H}}_{\mathrm{ag}}^{\mathrm{dec}})$;

18    $\mathbf{H}_{\mathrm{ab}}^{\mathrm{dec}(\ell)} \leftarrow \mathbf{H}_{\mathrm{ab}}^{\mathrm{dec}(\ell-1)} + \mathrm{FFN}(\widetilde{\mathbf{H}}_{\mathrm{ab}}^{\mathrm{dec}})$;

19 **end**

    // **Bipartite Interaction Prediction**

20 Compute bidirectional attention scores:;

21 $\mathbf{S}_{\mathrm{ag}\rightarrow\mathrm{ab}} \leftarrow \dfrac{(\mathbf{H}_{\mathrm{ag}}^{\mathrm{dec}} \mathbf{W}_Q^{\mathrm{out}})(\mathbf{H}_{\mathrm{ab}}^{\mathrm{dec}} \mathbf{W}_K^{\mathrm{out}})^{\top}}{\sqrt{d_k}}$;

22 $\mathbf{S}_{\mathrm{ab}\rightarrow\mathrm{ag}} \leftarrow \dfrac{(\mathbf{H}_{\mathrm{ab}}^{\mathrm{dec}} \mathbf{W}_Q'^{\mathrm{out}})(\mathbf{H}_{\mathrm{ag}}^{\mathrm{dec}} \mathbf{W}_K'^{\mathrm{out}})^{\top}}{\sqrt{d_k}}$;

23 Fuse scores and apply sigmoid:;

24 $\mathbf{Z} \leftarrow \mathbf{w}^{\top}[\mathbf{S}_{\mathrm{ag}\rightarrow\mathrm{ab}} \, (\mathbf{S}_{\mathrm{ab}\rightarrow\mathrm{ag}})^{\top}] + b$;

25 $\hat{\mathcal{E}}_{\mathrm{bg}} \leftarrow \sigma(\mathbf{Z})$;

    // **Epitope Extraction**

26 Extract per-residue epitope probabilities via top-$k$ pooling:;

27 $(\hat{y}_{\mathrm{ag}})_i = \frac{1}{k} \sum_{j \in \text{top-}k(\hat{\mathcal{E}}_{\mathrm{bg}})_{i:}} (\hat{\mathcal{E}}_{\mathrm{bg}})_{ij}$;

28 **Function** MHCA $(Q, K, V)$:

29    $Q_h \leftarrow Q W_Q^h, K_h \leftarrow K W_K^h, V_h \leftarrow V W_V^h$;      // Project per head $h$

30    $\alpha_{ij}^h \leftarrow \mathrm{Softmax}_j\left(\dfrac{Q_{h,i} \cdot K_{h,j}^{\top}}{\sqrt{d_h}}\right)$;      // Attention scores

31    $C_i^h \leftarrow \sum_j \alpha_{ij}^h V_{h,j}$;      // Context vector

      **Result:** $\mathrm{Concat}(C^1, \ldots, C^H) W_O$

      ;      // Combine heads

32 **end**

33 **Function** FFN $(X)$:

34    $\hat{\mathcal{E}}_{\mathrm{bg}} \leftarrow \mathrm{SiLU}(X W_1 + b_1) W_2 + b_2$;      // $W_1 \in \mathbb{R}^{d \times d_{ff}}$, $W_2 \in \mathbb{R}^{d_{ff} \times d}$

      **Result:** $\hat{\mathcal{E}}_{\mathrm{bg}}$

35 **end**