# OpenReview forum: "EpiFormer: a transformer-based multi-relational equivariant graph neural network for antibody-aware epitope prediction"
_ICLR.cc/2026/Conference — ICLR 2026 Conference Withdrawn Submission_

### Official Review · Reviewer_T67f · 2025-10-29

**Soundness:** 3
**Presentation:** 2
**Contribution:** 2
**Rating:** 4
**Confidence:** 4

**Summary:**

This paper proposes a new model, Epiformer, for antigen epitope prediction. This model designs an E(3) GNN to process the protein graph and achieve competitive performance.

**Strengths:**

- The epitope prediction problem seems meaningful and interesting.
- The paper designs a good architecture, which sounds technical.
- The components of the model make sense and reasonable.

**Weaknesses:**

- The paper is not well-organized. (1) The Bipartite graph link prediction seems not to be a task in the paper, but it is formulated in the 3.1. (2) Sec. 2 summarizes all the related work together; it is better to have an explicit categorization. (3) For the losses, the core task is actually Node Classification Loss, but it puts the Edge Prediction Loss, Dice Loss for Graph Segmentation, and Per-Graph Sparsity Regularization together, which seems to the model needs to address those problems. Maybe it is better to merge them into the Auxiliary losses. (4) Some formulas and superscripts, and subscripts are too cumbersome; please consider making them succinct.
- Some important relevant work may be missed in the discussion (see next part)

**Questions:**

- GraphEPN[1] is also a recent work on epitope prediction; it is better to discuss it and even add it as a baseline if possible.
- For the loss coefficients, it is of very high precision. Can the authors conduct the sensitivity analysis of those parameters?
- Why does the model get poor Recall in Table 1?
- The model architecture contains the Equivarant network for monomers and uses cross attention for interaction between monomers. Many papers share a similar idea, such as DiffDock, DynamicBind, and NeuralMD. What do the authors think are the biggest advantages of the proposed model to extract the complex feature compared to other similar architectures?
- What about using the PPI prediction methods to address this problem?
[1]GraphEPN: A Deep Learning Framework for B-Cell Epitope Prediction Leveraging Graph Neural Networks

---

### Official Review · Reviewer_ou5U · 2025-10-30

**Soundness:** 2
**Presentation:** 3
**Contribution:** 2
**Rating:** 2
**Confidence:** 4

**Summary:**

The paper introduces EpiFormer, a novel encoder-decoder architecture for antibody-aware epitope prediction. The work aims to address several key challenges in the field, including the geometric complexity of protein interactions, severe class imbalance, data scarcity, and the need for antibody-specific modeling. The main contributions include the architectural innovation and a new loss function.

**Strengths:**

1. The design of EpiFormer by combining an E(3)-equivariant GNN for intra-chain geometry with a cross-attention mechanism for inter-chain context is a sound way to encode structure information and model interaction simultaneously.

2. The paper demonstrates a substantial performance gain over existing methods on a benchmark (AsEP). The authors validate each component of their proposed model by comprehensive ablation studies

3. The paper is clearly written, and the architecture is well-described.

**Weaknesses:**

1. My first concern is the motivation for this work. Although antibody-aware epitope prediction is an important question, given the rise of high-accuracy protein complex predictors, the motivation of this work needs to be enhanced.

2. Moreover, your method needs to be compared with complex structure prediction models like AlphaFold 3 or traditional protein docking methods like HDock.

3. In several works of protein structure modelling, the authors take the modules from AF2 (typically the IPA module) as the key components in the neural network. So the need for a new architecture in the community should be justified.

**Questions:**

See Weaknesses

---

### Official Review · Reviewer_wSvM · 2025-10-30

**Soundness:** 2
**Presentation:** 2
**Contribution:** 2
**Rating:** 2
**Confidence:** 4

**Summary:**

This paper presents EpiFormer, an encoder-decoder architecture for antibody-aware epitope prediction that combines E(3)-equivariant multi-relational graph neural networks with cross-attention mechanisms. The authors address the challenge of predicting epitopes by representing proteins as multi-relational graphs with four edge types and incorporating both geometric features and protein language model embeddings. The model employs parallel EGNN-R encoders for antigen and antibody chains with bidirectional cross-attention, followed by a decoder that predicts bipartite interaction matrices. A joint loss function combines edge reconstruction, node classification with class reweighting and Dice loss, and auxiliary inter-chain distance classification. EpiFormer achieves approximately 1.7x improvement over baselines in multiple classification metrics on the AsEP dataset.

**Strengths:**

**S1**. The adoption of multi-relational edges to capture distinct protein interaction types is conceptually straightforward yet empirically effective, as demonstrated by the ablation study in Table 4 showing that multi-relational graphs substantially outperform simple proximity-based graphs.

**S2**.  The experimental results demonstrate improvements over existing baselines across diverse metrics and challenging data splits.

**Weaknesses:**

**W1**. The authors claim in the abstract and introduction that current approaches suffer from *lack of sophisticated architectures to model complex interaction patterns* and rely on *ineffective protein representations, predominantly using sequence-based approaches*,  but these claims are not adequately justified in the context of modern protein structure prediction and binding site prediction methods. Specifically, the paper fails to discuss: (i) pre-trained protein foundation models such as AlphaFold2 [1], AlphaFold3 [2], ESMFold [3], Boltz-1 [4], and Boltz-2 [5] that could potentially be fine-tuned for epitope prediction and already incorporate sophisticated representations and attention mechanisms; (ii) established equivariant architectures specifically designed for proteins like Equiformer [6] and EquiformerV2 [7] that already address the geometric modeling concerns; (iii) recent binding site prediction methods that use structural representations rather than sequence-based approaches. The repeated emphasis on *standard GNN* limitations creates a strawman argument, as the field has moved well beyond standard message-passing networks to equivariant architectures, and this rhetorical framing weakens the paper's positioning of its actual contributions. A more honest framing would acknowledge the existing sophisticated methods and clearly articulate what specific aspects of antibody-aware epitope prediction remain challenging despite these advances.

**W2**. While the authors correctly identify *sparsity of known antigen-antibody complexes* as a fundamental limitation in the literature, the paper does not explain how EpiFormer specifically addresses this challenge beyond using a carefully designed loss function. The model architecture itself does not incorporate explicit mechanisms for few-shot learning, meta-learning, transfer learning from abundant general protein structures, or data augmentation strategies that would directly tackle data scarcity.

**W3**. The core architectural components, E(3)-equivariant message passing (EGNN [8]), multi-relational graph representations (GearNet [9]), and cross-attention mechanisms for protein complexes (ChepNet [10]), are individually well-established in the literature on protein-ligand binding and protein-protein interaction prediction. While the specific combination and application to antibody-aware epitope prediction represents a reasonable engineering contribution, the paper does not articulate what fundamental modeling insights or novel architectural principles emerge from this problem domain that would be of broader interest to the community.

**W4**. The joint loss function involves many hyperparameters, which raises concerns about the source of performance gains and generalization to other datasets. The paper lacks: (i) sensitivity analysis showing how performance varies as key hyperparameters are changed; (ii) analysis of whether the hyperparameter settings learned on the AsEP dataset would transfer to other antibody-antigen datasets or require extensive re-tuning.

**W5**. A fundamental baseline for antibody-aware epitope prediction is direct structural alignment: given a query antibody-antigen complex, computing pairwise $C_{\alpha}$ RMSD or interface RMSD between the query and reference antibody-antigen pairs to predict the binding site.

**W6**. The paper does not sufficiently articulate how epitope prediction alone enables or improves therapeutic antibody design workflows, nor does it provide case studies or validation with domain experts demonstrating practical impact. Specifically, in de novo antibody design, the antigen structure is known but the antibody does not yet exist, so *antibody-aware* epitope prediction seems inapplicable unless the method can work with designed or hypothetical antibody sequences. Without demonstrating how accurate epitope prediction translates to improved outcomes in actual antibody discovery, the practical significance remains unclear despite the strong benchmark performance.

---

**Reference**

[1] J. Jumper et al. *Highly accurate protein structure prediction with AlphaFold. Nature 2021.*

[2] J. Abramson et al. *Accurate structure prediction of biomolecular interactions with AlphaFold 3. Nature 2024.*

[3] Z. Lin et al. *Evolutionary-scale prediction of atomic-level protein structure with a language model. Science 2023.*

[4] J. Wohlwend et al. *Boltz-1 democratizing biomolecular interaction modeling. BioRxiv 2025.*

[5] S. Passaro et al. *Boltz-2: Towards accurate and efficient binding affinity prediction. BioRxiv 2025.*

[6] Y. Liao et al. *Equiformer: Equivariant Graph Attention Transformer for 3D Atomistic Graphs. NeurIPS 2022.*

[7] Y. Liao et al. *EquiformerV2: Improved Equivariant Transformer for Scaling to Higher-Degree Representations. ICLR 2024.*

[8] V. Satorras et al. *E(n) Equivariant Graph Neural Networks. ICML 2021.*

[9] Z. Zhang et al. *Protein Representation Learning by Geometric Structure Pretraining. ICLR 2023.*

[10] H. Lim et al. *CheapNet: Cross-attention on Hierarchical representations for Efficient protein-ligand binding Affinity Prediction. ICLR 2025.*

**Questions:**

**Q1**. Can general binding site prediction models be adapted to antibody-aware epitope prediction, and if so, how does EpiFormer compare?

**Q2**. How sensitive is performance to the choice of protein language model embeddings, and could more recent foundation models further improve results?

**Details Of Ethics Concerns:**

No concerns

---

### Note · Authors · 2025-11-14

I have read and agree with the venue's withdrawal policy on behalf of myself and my co-authors.